# Motionless volumetric photoacoustic microscopy with spatially invariant resolution

Jiamiao Yang[1,2], Lei Gong[2,3], Xiao Xu[2], Pengfei Hai[1,2], Yuecheng Shen (ID) [1,2], Yuta Suzuki[2] & Lihong V. Wang[1]

Photoacoustic microscopy (PAM) is uniquely positioned for biomedical applications because of its ability to visualize optical absorption contrast in vivo in three dimensions. Here we propose motionless volumetric spatially invariant resolution photoacoustic microscopy (SIR-PAM). To realize motionless volumetric imaging, SIR-PAM combines two-dimensional Fourier-spectrum optical excitation with single-element depth-resolved photoacoustic detection. To achieve spatially invariant lateral resolution, propagation-invariant sinusoidal fringes are generated by a digital micromirror device. Further, SIR-PAM achieves 1.5 times finer lateral resolution than conventional PAM. The superior performance was demonstrated in imaging both inanimate objects and animals in vivo with a resolution-invariant axial range of 1.8 mm, 33 times the depth of field of the conventional PAM counterpart. Our work opens new perspectives for PAM in biomedical sciences.

[1] Caltech Optical Imaging Laboratory, Andrew and Peggy Cherng Department of Medical Engineering, Department of Electrical Engineering, California Institute of Technology, Pasadena, California 91125, USA. [2] Optical Imaging Laboratory, Department of Biomedical Engineering, Washington University in St Louis, Campus Box 1097, One Brookings Drive, St Louis, Missouri 63130, USA. [3] Department of Optics and Optical Engineering, University of Science and Technology of China, Hefei, Anhui 230026, China. Jiamiao Yang and Lei Gong contributed equally to this work. Correspondence and requests for materials should be addressed to L.V.W. (email: LVW@caltech.edu)

Three-dimensional (3D) optical imaging technologies, such as confocal microscopy[1], multiphoton microscopy[2, 3], and optical coherence tomography[4, 5], play fundamental roles in biomedical research. Distinct from these technologies, photoacoustic (PA) microscopy (PAM), as an emerging technology, has drawn increasing interest because of its capability to directly sense optical absorption in vivo and to facilitate either label-free or labeled 3D visualization of tissue[6–10]. PAM has an inherent depth resolving ability, thanks to the time-of-flight information carried by the PA signals, which enables volumetric imaging with only two-dimensional (2D) raster scanning[8]. However, conventional PAM suffers a rapidly degrading lateral resolution with the distance from the focal plane: For a given wavelength, the focusing objective lens has a limited depth of field (DOF), which inevitably deteriorates the volumetric image quality. Besides, the mechanical raster scanning involved in conventional PAM requires compromises between imaging speed and stability, restricting its performance. In fact, many other 3D imaging modalities, such as interferometric microscopy[11], optical frequency-domain imaging[12], and optical coherence tomography[5, 13], suffer from the same problems.

Recently, 2D single-pixel Fourier-spectrum acquisition imaging has been proposed (Supplementary Note 1 and Supplementary Fig. 1), which is achieved by using sinusoidal fringes with varying frequencies and phases to acquire the spectrum of a 2D scene[14]. This imaging method improves the signal-to-noise ratio significantly through a multiplexed illumination by delivering more energy to the bucket detector. However, it has no depth resolving ability; further, its fringes suffer from a limited DOF[15] which is undesirable for volumetric imaging.

Here we propose motionless volumetric spatially invariant resolution photoacoustic microscopy (SIR-PAM), achieved through Fourier-spectrum optical excitation with simultaneous single-element photoacoustic detection at all depths. SIR-PAM further develops 2D single-pixel Fourier-spectrum acquisition imaging by overcoming the above-mentioned limitations. Propagation-invariant sinusoidal fringes (PISFs) are generated by

wavefront engineering to produce in-focus fringes at any depth. Further, the time-of-flight information of the PA signal enables resolving the spectra of the object's cross-sections at different depths with a spatially invariant resolution. By using a digital micromirror device (DMD) to digitally generate the PISFs, we built a motionless SIR-PAM prototype with a resolution-invariant axial range (RIAR) of 1.8 mm, which is 33 times the DOF of the conventional PAM counterpart. Additionally, the lateral resolution of our system is 1.5-fold finer than that of conventional PAM. Our SIR-PAM prototype's performance was demonstrated by 3D imaging of both non-biological and biological objects. The spatially invariant high resolution and motionless 3D image acquisition is expected to open up new opportunities for PAM in biomedical applications.

## Results

**Principle of SIR-PAM.** As illustrated in Fig. 1a, SIR-PAM uses a series of PISFs with different spatial frequencies and phases to stimulate PA signals within an object. By using the phase-shifting method and resolving the depth based on time-of-flight information carried by PA signals, we can extract the Fourier spectrum of the optical absorption distribution at each depth with a single-element ultrasonic transducer. Then, a volumetric PA image with SIR is reconstructed. To realize motionless volumetric imaging, a DMD-based complex field modulation method is proposed to generate the PISFs digitally (see Methods section and Supplementary Movie 1).

The normalized 3D intensity profile of a PISF can be expressed as: $I_\varphi(x, y, z) = \frac{1}{2}\left[\cos\left(2\pi f_x x + 2\pi f_y y + \varphi\right) + 1\right]$, where $f_x$ and $f_y$ are spatial frequencies, and $\varphi$ is a shifting phase. To obtain such an intensity distribution, we modulate the optical field on the focal plane of an objective as

$$E_{\text{mod}}(x, y, \varphi) = \cos\left(\pi f_x x + \pi f_y y + \varphi/2\right). \quad (1)$$

In angular spectrum representation, $E_{\text{mod}}(x, y, \varphi)$ can be decomposed into two symmetrical plane waves $E_1$ and $E_2$ with

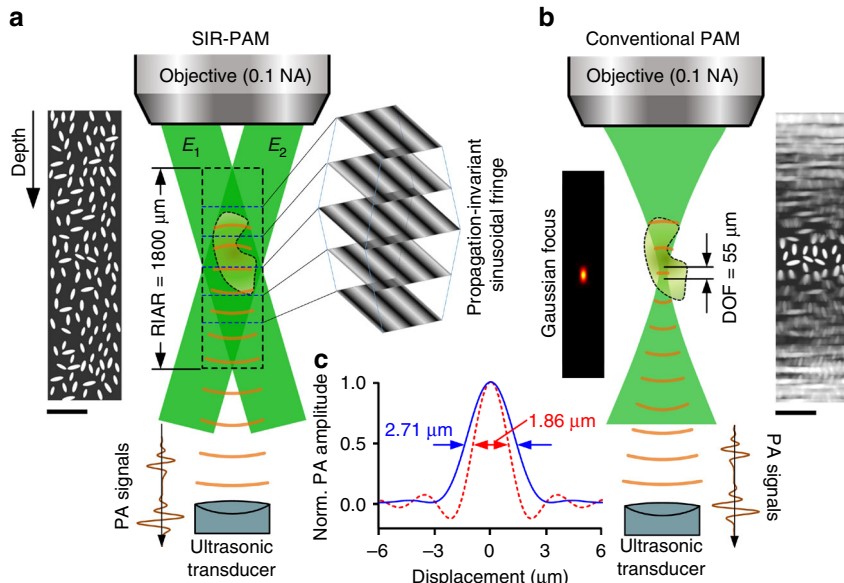

**Fig. 1** Principle of motionless SIR-PAM. **a** Principle of spatially invariant resolution photoacoustic microscopy (SIR-PAM). To achieve spatially invariant resolution volumetric imaging, SIR-PAM uses PISFs to stimulate PA signals within an object and extract its Fourier spectra, and uses an ultrasonic transducer to resolve the depth of the PA signals. **b** Principle of conventional PAM. A focused Gaussian beam is utilized in conventional PAM to stimulate PA signals, restricting its DOF. Besides, conventional PAM uses raster scanning to realize volumetric imaging. An object with uniformly distributed microparticles was simulated to compare the image qualities of two methods. **c** PSFs of SIR-PAM (*red curve*) and conventional PAM (*blue curve*), which indicate that SIR-PAM achieves 1.5 times finer lateral resolution than conventional PAM. *Scale bars*, 100 μm

expressions of $\frac{1}{2}\exp[i(\pi f_x x + \pi f_y y + \varphi/2)]$ and $\frac{1}{2}\exp[i(-\pi f_x x - \pi f_y y - \varphi/2)]$. Limited by the numerical aperture (NA) of the objective, $(\pi f_x)^2 + (\pi f_y)^2 < (\text{NA} \times 2\pi/\lambda)^2$, where $\lambda$ is the wavelength. In free space, the optical field formed by the propagation of $E_{\text{mod}}(x, y, \varphi)$ is $E_{\text{pro}}(x, y, z; \varphi) = \exp(ik_z z) \cos(\pi f_x x + \pi f_y y + \varphi/2)$, where $k_z$ is the axial wave vector of $E_1$ and $E_2$. Thus, the corresponding intensity $I_\varphi(x, y, z)$, which equals $\frac{1}{2}[\cos(2\pi f_x x + 2\pi f_y y + \varphi) + 1]$, is propagation invariant, that is, independent of $z$.

When a PISF with spatial frequencies of $f_x$ and $f_y$ illuminates an object, the modulated PA signals stimulated from each depth of the illumination volume are proportional to

$$S_\varphi(f_x, f_y, z) = \iint_{R(z)} \mu_a(x, y, z) \times I_\varphi(x, y, z) \mathrm{d}x\mathrm{d}y$$
$$= \frac{1}{2} \iint_{R(z)} \mu_a(x, y, z) \times [\cos(2\pi f_x x + 2\pi f_y y + \varphi) + 1] \mathrm{d}x\mathrm{d}y, \quad (2)$$

where $R(z)$ represents the illuminated area with intensity modulation at depth $z$, and $\mu_a(x, y, z)$ is the optical absorption distribution of the object. Thanks to the time-of-flight information carried by the received PA signals, the signal $S_\varphi(f_x, f_y, z)$ from each depth $z$ can be resolved. Then the corresponding Fourier coefficient at $z$ is extracted by using the phase-shifting method (Supplementary Note 2). Therefore, with the illumination of a certain frequency PISF, the corresponding Fourier coefficients of all of the cross-sections can be derived from the received PA signals. Combining the PISFs with all frequencies ($f_x$, $f_y$) in the Fourier domain allows us to retrieve accurately the Fourier coefficients of each cross-section. Finally, the volumetric PA image of the object is reconstructed layer-by-layer using inverse Fourier transformation of the Fourier coefficients. Benefiting from the PISFs, the modulation transfer functions at each depth are the same, enabling SIR-PAM spatially invariant resolution.

In practice, because the generated fringe has a size limited by the DMD and relay system[16], the fringe field can be regarded as the composition of two symmetrically collimated beams with a limited size. The RIAR of the SIR-PAM system is determined by the overlap of the two collimated beams, as illustrated in Fig. 1a. Besides, due to the limited range of PISFs, the resolution-invariant region in a transverse plane perpendicular to the optical axis decreases with the distance from the focal plane. Nevertheless, the achievable RIAR of SIR-PAM is much greater than the DOF of conventional PAM (Fig. 1b).

To compare quantitatively the lateral resolution of SIR-PAM with conventional PAM, we calculated its point spread function

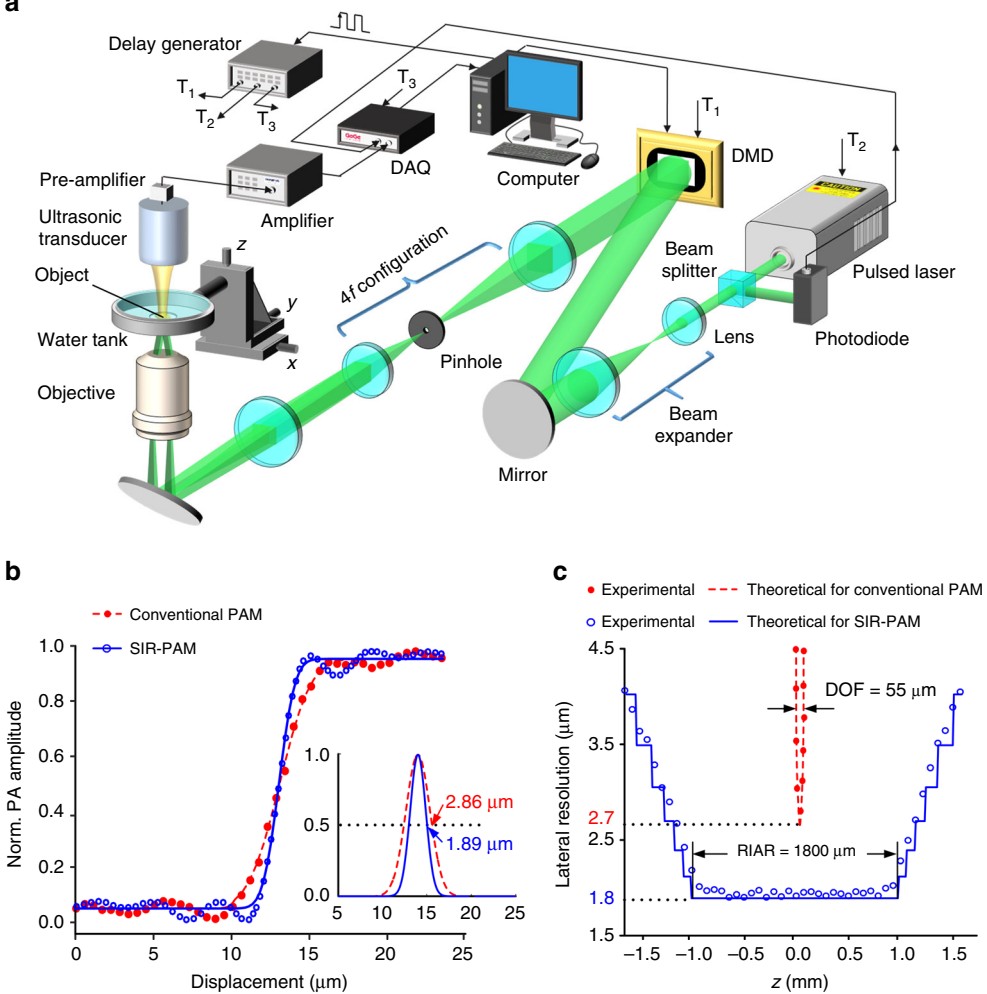

**Fig. 2** Experimental set-up and characterization of SIR-PAM system. **a** Schematic illustration of the SIR-PAM system. DAQ data acquisition system; DMD digital micromirror device; $T_1$ DMD trigger; $T_2$ pulsed laser trigger; $T_3$ DAQ trigger. **b** Normalized (Norm.) edge spread functions of conventional PAM and SIR-PAM, measured at the focal plane using a sharp edge made of deposited chromium. The corresponding line spread functions were fitted to compute the lateral resolutions defined by the full-width at half-maximum (*inset*). **c** Theoretical and experimental lateral resolutions of the two PAM systems versus depth. DOF depth of field; RIAR resolution-invariant axial range

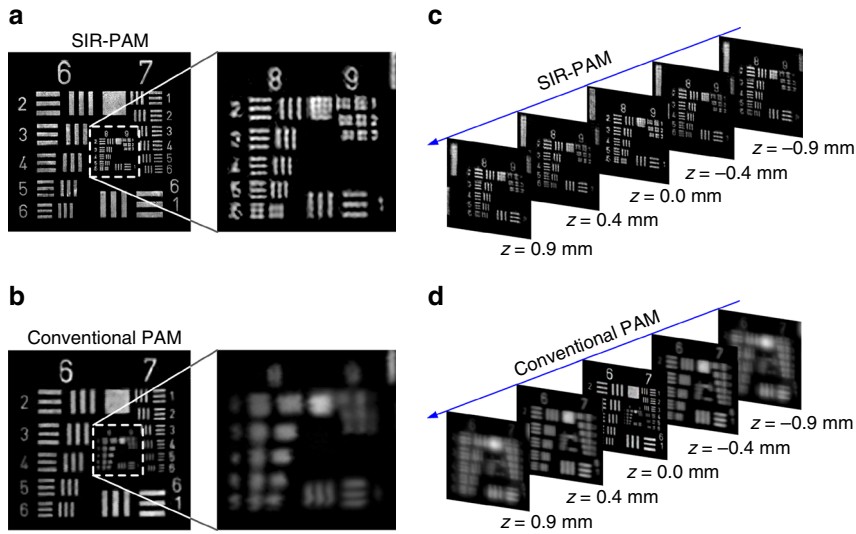

**Fig. 3** SIR-PAM imaging of a USAF resolution target with spatially invariant resolution and improved lateral resolution. Images of a resolution target (1951 USAF) acquired with SIR-PAM **a** and conventional PAM **b** at the focal plane, respectively. **c** Images of groups 8 and 9 acquired by SIR-PAM at different imaging depths. The focal plane is located at $z = 0.0$ mm. **d** Images of groups 6 and 7 acquired by conventional PAM at depths corresponding to **c**. The results show that SIR-PAM is capable of achieving spatially invariant resolution and improved lateral resolution

(PSF), which has the form of $\frac{2\pi^2 \mathrm{NA}}{\lambda r} J_1\left(\frac{4\pi r \mathrm{NA}}{\lambda}\right)$ in polar coordinates $(r, \theta)$ (Supplementary Note 3). The full-width at half-maximum (FWHM) of SIR-PAM's PSF is given by $0.35\lambda/\mathrm{NA}$, 1.5 times less than the FWHM of conventional PAM's PSF, given by $0.51\lambda/\mathrm{NA}$ (Fig. 1c and Supplementary Note 3). Therefore, SIR-PAM achieves a 1.5-fold finer lateral resolution than conventional PAM.

**Experimental set-up and characterization**. To realize motionless volumetric imaging, we introduce a technique based on wavefront engineering to generate the PISFs. By exploiting the ability of complex field modulation[17, 18] and the advantage of high switching rate[19], a single DMD enables high fidelity PISF generation and rapid switching among fringes with various frequencies (see Methods section and Supplementary Movie 1). Employing the flexible DMD-based scheme, we built an SIR-PAM prototype that is schematically illustrated in Fig. 2a (for a detailed description, see Methods section). The SIR-PAM system was characterized by imaging a sharp edge made of deposited chromium placed at different depths in water. As presented in Fig. 2b, the edge spread function was first measured at the focal plane, and its corresponding line spread function (LSF) was fitted to compute the lateral resolution. The lateral resolution, defined by the FWHM of the LSF, was 1.89 μm. In contrast, with the same objective, the conventional PAM system (Supplementary Note 4 and Supplementary Fig. 2) could achieve a lateral resolution of only 2.86 μm. In particular, the resolution of SIR-PAM system remains essentially unchanged (within 8% degradation) within an RIAR of 1800 μm (Fig. 2c), whereas the conventional PAM counterpart has a DOF of only 55 μm. Note that the axial resolutions of both systems are 19 μm, dictated by the bandwidth of the ultrasonic transducer[20].

The improved lateral resolution and spatially invariant resolution of SIR-PAM were directly verified by imaging a USAF resolution target placed at different depths. Figure 3a, b shows the reconstructed images acquired by SIR-PAM and conventional PAM at the focal plane ($z = 0$ mm), respectively. Sections of groups 8 and 9 resolved by the two imaging methods are enlarged for further examination. With SIR-PAM, the features of group 8, element 5 can be clearly resolved with a resolution of 406.4 line pairs per mm, whereas the smallest features that can be resolved

by conventional PAM are 256.0 line pairs per mm in resolution (in group 8, element 1).

Figure 3c, d shows images acquired at different imaging depths. For conventional PAM, the features of element 3 in group 6 (resolution of 80.6 line pairs per mm) are hardly resolved at an imaging depth of ± 0.4 mm; when the imaging depth reaches ± 0.9 mm, all the patterns in groups 6 and 7 (lowest resolution of 64.0 line pairs per mm) become blurred beyond recognition (Fig. 3d). In contrast, SIR-PAM can resolve the features of element 5 in group 8 (resolution of 406.4 line pairs per mm) wherever the resolution target is located in the range from −0.9 to 0.9 mm, as shown in Fig. 3c. Thus, SIR-PAM has a much larger RIAR with finer resolution than conventional PAM.

**Volumetric imaging of an inanimate object**. To demonstrate the volumetric imaging performance of our SIR-PAM system in comparison with a conventional PAM system, we imaged a sharp-featured 3D object using both systems over a volume of 1.2 mm × 0.9 mm (lateral) × 1.8 mm (axial). The 3D object was made of spatially distributed carbon fibers (about 7 μm diameter) fixed in gelatin. Figure 4a, b and the animation in Supplementary Movie 2 show the volume-rendered images. As expected, SIR-PAM provides high-resolution imaging throughout the entire volume, whereas conventional PAM provides high resolution only in the limited focal region. Three en face image slices were taken at different depths to compare the two systems in image quality. The corresponding line profiles across the carbon fibers are shown in Fig. 4c−e. Toward the two ends of the depth range, the crossed fibers remain clearly resolvable in the SIR-PAM images but become grossly blurred in the conventional PAM images. Even on the focal plane, SIR-PAM yields a sharper image.

**SIR-PAM imaging of zebrafish larvae in vivo**. We further applied SIR-PAM to noninvasively image zebrafish larvae in vivo. Several 3-day-old wild-type zebrafish embryos were carefully mounted on angled slide glasses and placed in a water tank to keep them alive. The tilted mount allowed imaging over a depth range of 1.8 mm (see Methods section). Whole-body imaging of living zebrafish larvae was performed with SIR-PAM and conventional PAM. Figure 5a, b depicts the reconstructed images of

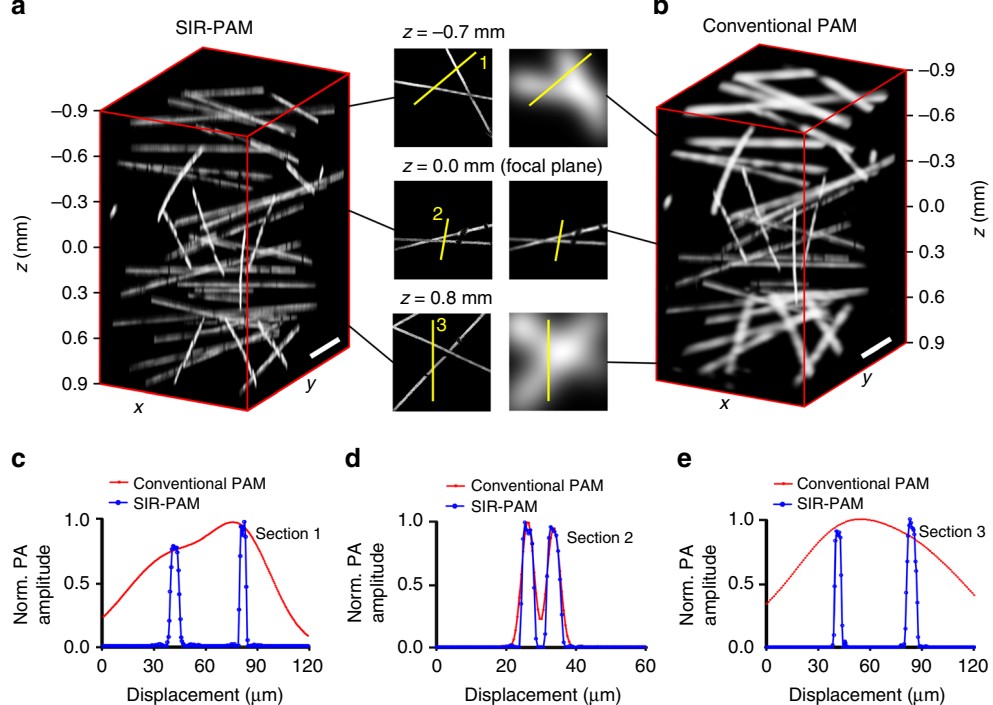

**Fig. 4** Volumetric imaging of spatially distributed carbon fibers. **a** Volume-rendered image of a 3D object reconstructed by SIR-PAM, showing spatially invariant resolution. **b** Image of the same region as in **a** obtained by conventional PAM with decreasing lateral resolution away from the focal plane. Three pairs of en face image slices were taken at imaging depths of −0.7, 0.0, and 0.8 mm. The focal plane is located at $z = 0.0$ mm. **c**−**e** Corresponding line profiles across the carbon fibers shown to compare the imaging quality of the two systems. *Scale bars*, 150 μm

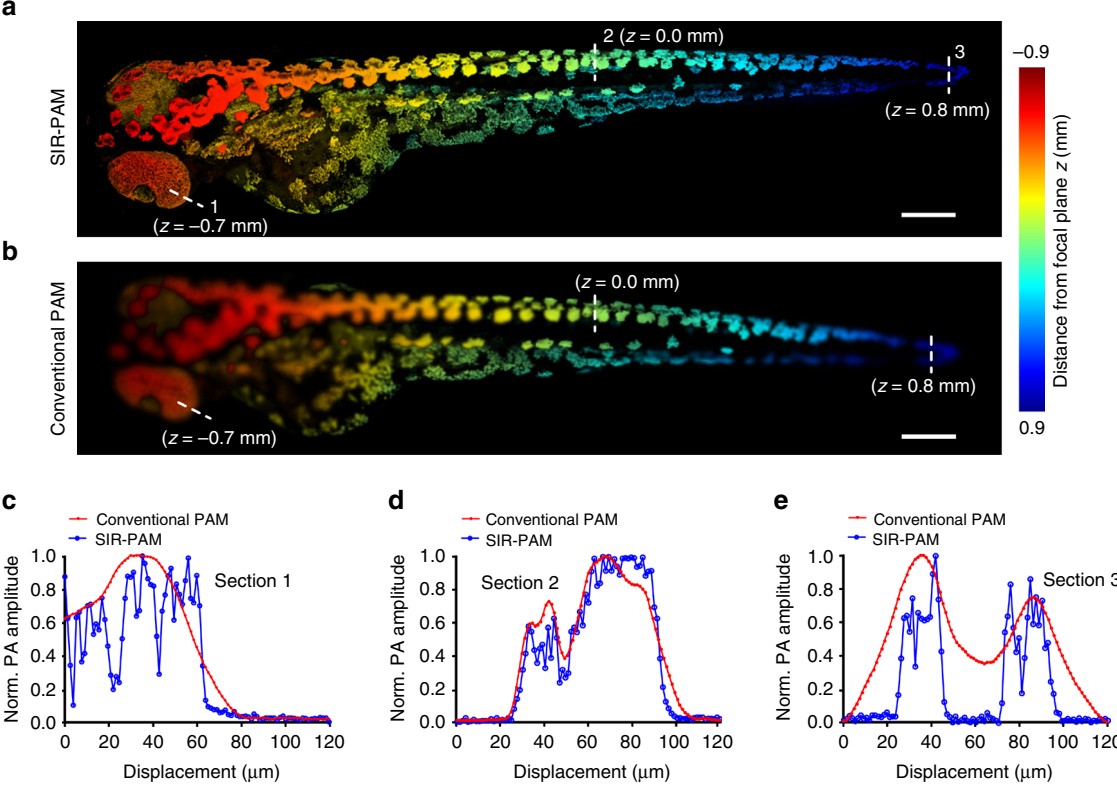

**Fig. 5** SIR-PAM imaging of zebrafish embryos in vivo. **a**, **b** Depth-encoded whole-body images of a zebrafish larva obtained by SIR-PAM **a** and conventional PAM **b**. The focal plane is located at $z = 0.0$ mm. **c**−**e** Normalized PA amplitude profiles along the dashed lines. Finer structures can be resolved by SIR-PAM due to its improved lateral resolution within a large RIAR. *Scale bars*, 200 μm

the whole zebrafish larva, with imaging depths encoded in color. An animation (Supplementary Movie 3) shows all the image slices taken at different depths and the corresponding enlarged view of the localized structures. It is obvious that the SIR-PAM images are much superior in resolution at all depths. Using this technique, structures throughout the whole fish can be clearly resolved. In contrast, the images obtained by conventional PAM blur quickly away from the focal plane and become much blurred toward the two ends of the depth range.

In addition, fine structures of the fish at different imaging depths were examined and the corresponding line profiles across these structures are presented in Fig. 5c–e. SIR-PAM resolves more fine structures than conventional PAM within the fish's eye, 0.7 mm above the focal plane (Fig. 5c), and within the fish's tail, 0.8 mm below the focal plane (Fig. 5e). Even on the focal plane, the edges of the imaged structures acquired by SIR-PAM are sharper (Fig. 5d).

Note that, during the imaging experiments, the averaged laser intensity on the fish surface was about 189 mW cm$^{-2}$, below the 200 mW cm$^{-2}$ safety limit set by the American National Standards Institute. Thus, our observations confirm that non-invasive, whole-body, and in vivo 3D imaging of zebrafish larvae with a spatially invariant resolution over a large depth range has been achieved by SIR-PAM. This technique paves the way to study various embryos or larvae of other animal models in vivo.

## Discussion

In conclusion, we have proposed and implemented SIR-PAM by adopting PISFs and Fourier-spectrum acquisition at all photo-acoustically resolved depths. In particular, SIR-PAM is capable of direct 3D imaging of an object without mechanical scanning. A versatile SIR-PAM system based on a DMD was built. For both inanimate objects and in vivo animals, our system demonstrated a 1.8 mm RIAR, which is 33 times the DOF of the conventional PAM counterpart, as well as a 1.5-fold enhancement in lateral resolution.

SIR-PAM can be further improved. With the current system, the field of view (FOV) is limited, and a larger image is obtained via montage. According to Supplementary Note 5 and Supplementary Fig. 3, the product of the FOV and effective numerical aperture is restricted by the pixel count of the DMD (Supplementary Eq. 7). Therefore, there is a tradeoff between the FOV and the lateral resolution, which is determined by the numerical aperture. However, we can improve them simultaneously with a higher-resolution DMD. Increasing the pixel count also extends the RIAR (Supplementary Eq. 8). In addition, because the axial resolution of SIR-PAM is determined by the bandwidth of the ultrasonic transducer, a wider bandwidth can be utilized to achieve a higher axial resolution. With the same pulse repetition rate, the scan time of SIR-PAM is about 1.2 times longer than conventional PAM for the same imaged region. Currently, the imaging speed is mainly limited by the pulse repetition rate (1 kHz) of the laser, at which rate it took about 21 s to acquire a volumetric image within the DMD-defined FOV. The acquisition time can be shortened to 1 s by increasing the laser pulse repetition rate to the full switching rate (22.7 kHz) of the DMD. The speed can be further improved by compressed sensing[21, 22] for potentially video-rate volumetric imaging. Moreover, the broad operational spectrum ($\lambda = 400$–2500 nm) of the DMD could facilitate wide spectral imaging of the optical absorption of biological tissues in vivo[6, 23] with the same set-up.

In the experiments described above, only slightly scattering media, such as scattering gel or zebrafish embryos, were imaged. Strong scattering adversely affects the performance of SIR-PAM. To investigate the effect of scattering on SIR-PAM, we imaged

tissue-mimicking phantoms with different scattering coefficients (Supplementary Note 6 and Supplementary Figs. 4 and 5). SIR-PAM was found to be more susceptible to scattering than conventional PAM. To advance the penetration of SIR-PAM, one could consider optical clearing[24, 25] (Supplementary Note 7 and Supplementary Fig. 6), which could be suited for selected applications.

Overall, our work brings a new step change to PAM technologies and will motivate studies in biology and medicine where in vivo 3D imaging with spatially invariant resolution is desirable. For instance, SIR-PAM can enable high-resolution functional imaging of angiogenesis, melanoma, and hemoglobin oxygen saturation ($sO_2$) within the skin and other superficial organs by measuring optical absorption contrasts. Its motionless volumetric imaging ability paves the way to whole-body study of animal embryos in vivo. Furthermore, in combination with optical clearing[25], SIR-PAM can provide deeper imaging. Apart from providing the new perspective for PAM, the DMD-based PISF illumination method may be further exploited for structured-illumination optical microscopy[26, 27] and 3D single-pixel imaging[28, 29].

## Methods

**Propagation-invariant sinusoidal fringes generated by a DMD.** A binary DMD was used to generate the PISFs, which required spatially encoding the complex field (including the amplitude and phase) of a light beam with binary holograms. Here we employed a super-pixel encoding method[17, 18] to design the required holograms. In this method, the square regions of nearby pixels ($4 \times 4$ pixels within $768 \times 768$ pixels in our case) were grouped into various super-pixels to define a complex field in the imaging plane, using the first-order diffraction beam. Practically, encoding was achieved by applying a sequence of ON and OFF states to the micromirrors of the DMD that were in the optical path of each user-defined region of interest. The high fidelity of this method allows accurate generation of the desired fields. Experimentally, super-pixel based field modulation was realized by a 4f configuration with the DMD and a low-pass filter[14, 17] (Fig. 2a).

First, a binary DMD hologram was calculated with the super-pixel encoding method, according to the phase pattern and normalized amplitude of each PISF's optical field (Supplementary Fig. 7a–c). Then the special optical field was produced in the imaging plane when the hologram was loaded onto the DMD. Supplementary Fig. 7d presents the intensity distribution of the generated field on a given plane, which agrees with the theoretical distribution. In the same way, other PISF's optical field could be created with their corresponding holograms projected. Finally, the high-speed switching ability of the DMD enabled us to vary PISFs rapidly with different spatial frequencies. During this process, we could directly observe two scanning collimated beams, which are vividly displayed in Supplementary Movie 1. Our DMD-based scheme enables generation and high-speed switching of PISFs, enabling motionless volumetric imaging.

**Details of the experimental set-up.** As shown in Fig. 2a, a laser (532 nm wavelength; Elforlight, Ltd.) with a 1 kHz pulse repetition rate and a 5 ns pulse duration was used as the excitation light source. The pulse energy was monitored by a photodiode detector (SM05PD1A, Thorlabs, Inc.) to compensate for energy fluctuations. A beam expander with a 20 times magnification was used to enlarge the laser beam. A DMD ($1024 \times 768$ pixel resolution; Texas Instruments, Inc.) was used to generate PISFs by complex field encoding. Here, 21303 PISFs were examined to acquire the Fourier coefficients (Supplementary Note 8), then the generated fringes were relayed by a converging lens ($f = 60$ mm) and an objective lens (NA = 0.1; Olympus, Corp.) to stimulate PA waves within an object. To detect the PA signals, a focused ultrasonic transducer (50 MHz central frequency, 70% bandwidth, 6 mm element diameter, 48 mm focal length, and 3.8 mm DOF; V358-SU, Olympus, Corp.) was placed on the other side of the object confocally with the objective lens and then coupled by water. The PA signals were amplified by two electronic amplifiers with gains of 24 and 30 dB, respectively. The amplified signals were acquired by a data acquisition system (DAQ, Razor 14, GaGe, Corp.) controlled by a computer. The computer was also used to synchronize the pulsed laser, the DMD, and the DAQ via a delay generator (see the time sequence diagram in Supplementary Fig. 8). The FOV dictated by the effective illumination area was $180 \times 180$ μm$^2$, which was covered by the focal diameter of the focused transducer. To obtain a larger image, a montage strategy based on movement of the object with a step size of 170 μm was adopted.

**Zebrafish preparation for imaging experiments.** The wild-type zebrafish embryos were raised and cared for by the zebrafish facility at Washington University in St. Louis. We were authorized to handle the living fish under a

recently developed protocol (Protocol Number: 20160109). All experimental animal procedures were carried out in conformity with the laboratory animal protocol approved by the Animal Studies Committee of Washington University in St. Louis. Several 3-day-old embryos were imaged in our experiments. Before the experiments, the zebrafish larvae were kept in saline water at 28.5 ℃ and anaesthetized using (0.5–2.0 mg ml$^{-1}$) tricaine (MS-222, Western Chemical, Inc.)[30]. Next, they were carefully moved and placed on angled slide glasses using a pipette, so that tilted fish with an extended depth range were prepared for imaging. The fish were immobilized with melted agarose (2%; Sigma-Aldrich, Corp.). Then the slide glass was moved into a plastic culture dish that served as a water tank for acoustic coupling. This preparation minimized the movement of the zebrafish larvae during the experiments, while keeping them alive.

**Data availability**. The data that support the findings of this study are available from the authors on reasonable request, see author contributions for specifics.

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

## Acknowledgements
We thank Dr Yan Liu, Toru Imai, Dr Jinyang Liang, Dr Cheng Ma, and Ashton Hemphill for insightful discussions on the work. L. Gong acknowledges the support from China Scholarship Council (Grant No. 201506340017). The authors appreciate Prof. James Ballard's critical reading of the manuscript. This work was supported by the National Institutes of Health (NIH) grants DP1 EB016986 (NIH Director's Pioneer Award) and R01 CA186567 (NIH Director's Transformative Research Award).

## Author contributions
L.V.W. conceived the project. J.Y. and L.G. designed the research, built the system, and performed the experiments. J.Y. wrote the codes for the experiments. X.X. contributed to the system design and manuscript preparation. P.H. performed conventional PAM experiments. J.Y. and Y.Sh. performed PAM experiments in scattering media. Y.Su. performed precursor experiments. J.Y. and L.G. analyzed the experimental results and wrote the manuscript. L.V.W. provided overall supervision. All authors were involved in revising the manuscript.

## Additional information

**Competing interests:** L.V.W. has a financial interest in Microphotoacoustics, Inc., which, however, did not support this work. The remaining authors declare no competing financial interests.

