## [Peer Review File · Nature Communications]

Reviewers' comments:

Reviewer #1 (Remarks to the Author):

The authors describe a technique to expand the spatial resolution in photoacoustic microscopy. The technique utilizes previously published approaches, i.e., <http://www.nature.com/articles/srep01116>, but with important new advantages for photoacoustic imaging. The images are very impressive and clearly illustrate the advantage of this approach over conventional PAM.

1. The authors should compare the scan time including processing time between the two techniques.
2. The increase in DOF and lateral resolution are impressive. How do these metrics change as a function of objective and transducer center frequency?
3. One wonders about the word choice of motionless. If a single element transducer is used, then there must be some type of motion to create an image, correct?
4. Some additional details on triggering the Laser, DAQ, and DMD would be useful.
5. In the experimental setup in Figure 2, the authors show an amplifier to increase the photoacoustic amplitude as well as a beam expander but there are no details about either in the manuscript. More details are needed here including the gain used in the amplifier.
6. Wavefront engineering devices like SLM and DMD can be sensitive to light source polarization, and polarizing beam splitters are often used with wavefront setups. What precautions were taken here?

Reviewer #2 (Remarks to the Author):

In this manuscript, the authors propose a method of motionless volumetric spatially invariant resolution photoacoustic microscopy (SIR-PAM). In their setup, a digital micro-mirror device was used to generate propagation-invariant sinusoidal fringes, to form spatially invariant lateral resolution. Also, a 2D Fourier-spectrum optical excitation along with a single-element depth-resolved photoacoustic detection was implemented to achieve motionless imaging. The authors demonstrated this imaging technique by 3D imaging of both non-biological and biological objects without mechanical scanning, with 1.5 times finer lateral resolution and 33 times depth of focus than conventional PAM.

Overall, the study reported here is important for PAM technology since it shows great improvement on the imaging depth of focus, which opens new perspectives for the applications of PAM, especially in transparent medium. It is well-written and along with convincing experiment results. However, there are some clarifications needed before it is considered for publication.

Comments:

1. Recently, Liang etc. has published two papers on photoacoustic microscopy using digital micromirror device ("Spatially Fourier-encoded photoacoustic microscopy using a digital micromirror device", *Opt Lett.* 2014 February 1, 39(3): 430–433; "Random-access optical-resolution photoacoustic microscopy using a digital micromirror device", *Opt Lett.* 2013 August 1, 38(15): 2683–2686). In their work, they developed spatially Fourier-encoded photoacoustic microscopy using a digital micromirror device. The latter one was also based on super-pixel on the DMD. The authors should compare the proposed system with Liang's previous work to demonstrate the main difference.
2. The authors claimed resolution axial range of 1.88mm, 33 times the depth of the conventional PAM, which is a large improve. Due to the optical scattering in tissue, the penetration depth of optical resolution photoacoustic microscopy in tissue is limited to ~ 1 mm. It would be helpful if the authors could make a discussion on the impact of their work regarding applications in this field.

3. In this manuscript, the authors claimed a resolution-invariant axial range of 1.8mm. What is the corresponding FOV with invariant resolution at +/- 0.9mm depth? This data may help to make a fair comparison between the proposed technology and conventional PAM. It is also preferable to include a brief description on the conventional PAM setup used here for comparison.

4. Line 122 on page 7, the authors claimed a resolution-invariant axial range of 1.8mm, within 8% degradation. What is the DOF defined by the 50MHz focused transducer used? It may not affect the resolution of the proposed system which is based on optical resolution. But what is the difference if using an unfocused transducer instead of a focused transducer? How does it affect the proposed system? SNR etc.

5. Line 98 on page 5, the authors derived the FWHM formula, which results in the FWHM of 1.86 μ m and 2.71 μ m for the proposed system and conventional PAM respectively. However, at line 129 and line 130, the resolution of proposed system and conventional PAM were 1.23 μ m and 1.95 μ m, respectively, less than the calculated value. Please comment on this.

6. Line 165 on page 9, the authors present the laser intensity on the fish surface was ~189mW/cm². Usually for the pulsed laser, 20mJ/cm² is used as ANSI limit. Is the ~189mW/cm² mentioned here based on the peak pulse energy?

Reviewer #3 (Remarks to the Author):

In this paper the authors make use of structured illumination approaches to generate propagation invariant sinusoidal functions for imaging in photoacoustic microscopy. This approach generates a depth-invariant lateral resolution for quite a large imaging depth, thus overcoming one of the drawbacks of traditional PAM. The authors present the basis for the generation of the depth-invariant functions and experimental results both on gel and zebrafish.

Overall, the paper is very interesting and the method shows great potential. However, the authors have not discussed a key factor when imaging in-vivo, which is the presence of scattering. How does depth-invariance change when imaging in a slightly scattering medium? The equation the authors propose after Eq(1) includes the transfer function $\exp(ikz)$. This transfer function in the presence of scattering has a real and a complex part, thus attenuating higher frequencies more than lower frequencies, changing the depth-invariance. How is this affected in practice? Is it negligible when dealing with tissues?

In this paper the authors present results of two cases which present very little scattering: a scattering gel, as appropriate to establish the resolution of the method, and a 3 day zebrafish embryo, which is one of the most used species in optical microscopy due to its transparency. What is missing is a thorough study of the effect of scattering on this new approach, and how it compares with traditional PAM. It is my feeling that traditional PAM may be less sensitive to scattering than SIR-PAM, but I might be mistaken. I believe this article would have a much stronger impact if a study of the effect of scattering is included, instead of the simple gel phantom. I would suggest the authors include equivalent measurements in gels which have different values of scattering, starting at 0 (the one measured) and going up to 10cm⁻¹ or nearby, comparing PAM vs SIR-PAM as done so far. This way the true potential of SIR-PAM will be well established and indeed would be a very important contribution to the community. I therefore would be happy to accept this paper after this major revision has been included.

Point-by-point responses to the reviewers' comments

We appreciate the efforts of all the reviewers in providing their insightful comments and valuable suggestions, which we have fully incorporated when revising our manuscript. Furthermore, we have performed additional experiments to address the questions the reviewers raised. The experimental details and results are presented in the supplementary materials, as well as some new figures. All the changes are highlighted in yellow in the revised version. Below are the detailed replies to the reviewers' comments.

Reviewer #1:

The authors describe a technique to expand the spatial resolution in photoacoustic microscopy. The technique utilizes previously published approaches, i.e., <http://www.nature.com/articles/srep01116>, but with important new advantages for photoacoustic imaging. The images are very impressive and clearly illustrate the advantage of this approach over conventional PAM.

[Reply]: We greatly appreciate your highly positive comments.

1.1. *The authors should compare the scan time including processing time between the two techniques.*

[Reply]: Thank you for the suggestion. For the current SIR-PAM and conventional PAM systems, a pulsed laser with a 1-kHz pulse repetition rate was used as the light source. The scanning step we set for conventional PAM is 1.35 μm , so its scan time corresponding to an imaged region of $180 \times 180 \mu\text{m}^2$ is $180 \times 180 / (1.35 \times 1.35) / 1000 = 17.8$ seconds, and it supports real-time data processing. For the SIR-PAM system, the total number of PISFs required is 21303 (Supplementary Note 4). Thus, the acquisition time is $21303 / 1000 = 21.3$ seconds, and the post data processing time is ~ 0.5 seconds. Thus, the total time cost of SIR-PAM is about 22 seconds, 1.2 times longer than that of conventional PAM. Accordingly, we have added the scan time comparison in the discussion section of the revised manuscript as follows.

“With the same pulse repetition rate, the scan time of SIR-PAM is about 1.2 times longer than conventional PAM for the same imaged region.” (Lines 195 – 196, Page 10).

1.2. *The increase in DOF and lateral resolution are impressive. How do these metrics change as a function of objective and transducer center frequency?*

[Reply]: Thank you for asking for this clarification. As illustrated in Fig. 1(a), the DOF of the SIR-PAM system (we called resolution-invariant axial range, RIAR) is determined by the overlap of the two collimated beams, whose incident angle is limited by the NA of the objective. According to Eq. (S.10) in Supplementary Note 6, if the pixel number of the DMD is fixed, the DOF will decrease with an increase in the objective NA. Furthermore, the lateral resolution of our system is defined as the full width at half maximum (FWHM) of the SIR-PAM's PSF, given by $0.35\lambda/\text{NA}$, which has been derived in Supplementary Note 3. Thus, the lateral resolution will be finer with an increase of objective NA. Actually, the DOF and lateral resolution are not related to the transducer center frequency that determines the axial resolution of both SIR-PAM and conventional PAM systems.

1.3. *One wonders about the word choice of motionless. If a single element transducer is used, then there must be some type of motion to create an image, correct?*

[Reply]: Thank you for the comment. In the SIR-PAM system, the whole field of view (FOV, $180\times 180\ \mu\text{m}^2$) is covered by the focal diameter of the focused transducer ($200\ \mu\text{m}$ in diameter). To obtain a PA image with a single-element transducer, SIR-PAM employs a series of propagation invariant sinusoidal patterns with different frequencies to acquire the spectrum of the object at each depth. Thus, only switching among the sinusoidal fringes, which is achieved by a programmable DMD, is required during the imaging process, eliminating the mechanical motion of both transducer and object. Therefore, this imaging process is motionless unlike conventional PAM, improving the stability of the imaging system.

To make this point clearer, some sentences have been revised in the manuscript: “we can extract the Fourier spectrum of the optical absorption distribution at each depth with a single-element ultrasonic transducer.” (Lines 63 – 64, Page 4); “The field of view (FOV) dictated by the effective illumination area was $180\times 180\ \mu\text{m}^2$, which was covered by the focal diameter of the focused transducer.” (Lines 122 – 123, Page 7).

1.4. *Some additional details on triggering the Laser, DAQ, and DMD would be useful.*

[Reply]: Thanks for your suggestion. Actually, a delay generator was adopted for the synchronization of the pulsed laser, DAQ, and DMD. Accordingly, the delay generator with trigger signals has been added in Fig. 2(a), and we have replaced the old figure with the new one in the

paper. Further, a time sequence diagram (Fig. S3) demonstrating the synchronization details is also added to the supplementary materials. In the manuscript, we revised the corresponding sentence as follows:

“The computer was also used to synchronize the pulsed laser, the DMD, and the DAQ via a delay generator (see the time sequence diagram in Fig. S3).” (Lines 120 – 122, Page 7)

Figure 2| (a) Schematic illustration of the SIR-PAM system.

Figure S3| Time sequence diagram for triggers of the DMD, laser, and DAQ synchronization.

To guarantee the pattern has been refreshed on the DMD when the pulsed laser illuminates, the laser trigger T2 is delayed for 50 μ s with respect to the DMD trigger T1. The DAQ trigger T3 is

synchronized to T2 with no delay to guarantee synchronous data acquisition from the ultrasonic transducer and the photodiode detector.

1.5. *In the experimental setup in Figure 2, the authors show an amplifier to increase the photoacoustic amplitude as well as a beam expander but there are no details about either in the manuscript. More details are needed here including the gain used in the amplifier.*

[Reply]: Yes, we agree and thank you for the suggestion. In the experimental setup, the magnification of the beam expander is 20 times. Actually, two amplifiers were adopted for the PA signal amplification. The gain of the pre-amplifier is 24 dB, and that of the other is 30 dB. These details have been added in the experimental setup description of the revised manuscript as follows:

“A beam expander with a 20-time magnification was used to enlarge the laser beam.” (Line 111, Page 6); “The PA signals were amplified by two electronic amplifiers with gains of 24 dB and 30 dB, respectively.” (Line 119, Page 7).

1.6. *Wavefront engineering devices like SLM and DMD can be sensitive to light source polarization, and polarizing beam splitters are often used with wavefront setups. What precautions were taken here?*

[Reply]: Indeed, the liquid crystal spatial light modulator (LC-SLM) is sensitive to the light source polarization because it uses the optical and electrical anisotropy of liquid crystal materials. Thus, a definite incident polarization is required for LC-SLM based wavefront shaping. On the other hand, the DMD is actually an array of reflective micro-mirrors whose surfaces are coated with aluminum thin film. It has been experimentally demonstrated [Yu-Xuan Ren et al., Appl. Opt. 49, 1838-1844 (2010)] that the aluminum coatings, which are commonly used in commercial DMDs, cause approximately 90° polarization rotation of the modulated beam compared to the incident polarization state. However, the DMD doesn't require a certain incident polarization state, and DMD enables modulating the laser beam with arbitrary polarization. Notably, the DMD is characterized by its polarization insensitivity, broad operational spectrum, and fast switching capability. These are the reasons why we choose the DMD to generate the propagation invariant sinusoidal beams for PA wave excitation in the SIR-PAM system.

Reviewer #2:

In this manuscript, the authors propose a method of motionless volumetric spatially invariant resolution photoacoustic microscopy (SIR-PAM). In their setup, a digital micro-mirror device was used to generate propagation-invariant sinusoidal fringes, to form spatially invariant lateral resolution. Also, a 2D Fourier-spectrum optical excitation along with a single-element depth-resolved photoacoustic detection was implemented to achieve motionless imaging. The authors demonstrated this imaging technique by 3D imaging of both non-biological and biological objects without mechanical scanning, with 1.5 times finer lateral resolution and 33 times depth of focus than conventional PAM.

Overall, the study reported here is important for PAM technology since it shows great improvement on the imaging depth of focus, which opens new perspectives for the applications of PAM, especially in transparent medium. It is well-written and along with convincing experiment results. However, there are some clarifications needed before it is considered for publication.

[Reply]: We truly appreciate your favorable comments on our new technology.

Comments:

2.1. *Recently, Liang etc. has published two papers on photoacoustic microscopy using digital micromirror device (“Spatially Fourier-encoded photoacoustic microscopy using a digital micromirror device”, Opt Lett. 2014 February 1, 39(3): 430–433; “Random-access optical-resolution photoacoustic microscopy using a digital micromirror device”, Opt Lett. 2013 August 1, 38(15): 2683–2686). In their work, they developed spatially Fourier-encoded photoacoustic microscopy using a digital micromirror device. The latter one was also based on super-pixel on the DMD. The authors should compare the proposed system with Liang’s previous work to demonstrate the main difference.*

[Reply]: Thank you for your suggestion. We are well aware of their work, in which they used a DMD to perform structured illumination, and employed a super-pixel to achieve alterable intensity via pixel averaging. The DMD was used as an intensity modulator, by which they could obtain intensity fringes with an accurate sinusoidal profile only at the focal plane. However, the intensity fringes suffered from the defocusing effect, yielding a very limited DOF. Thus, like conventional PAM, the spatially Fourier-encoded PAM they proposed also suffers a rapidly degrading lateral resolution with the distance from the focal plane. Taking a different approach, our work adopts a

super-pixel encoding method to achieve complex wavefront shaping, including the amplitude and phase of the optical field. So, the DMD here is used as a complex wavefront modulator, by which propagation invariant sinusoidal fringes (PISFs) are generated. The generated PISFs keep their accurate sinusoidal distribution over a long axial range. Based on these distinct PISFs, SIR-PAM enables us to achieve an extended DOF and enhanced lateral resolution.

According to your suggestions, we have cited Liang et al. as a reference in the Introduction and pointed out the limitation of their approach, as follows: “Nevertheless, fringes generated by traditional methods suffer from a very limited DOF¹⁶.” (Lines 43 – 44, Page 3)

16 Liang, J., Gao, L., Li, C. & Wang, L. V. Spatially Fourier-encoded photoacoustic microscopy using a digital micromirror device. *Opt. Lett.* 39, 430-433 (2014).

2.2. *The authors claimed resolution axial range of 1.88mm, 33 times the depth of the conventional PAM, which is a large improve. Due to the optical scattering in tissue, the penetration depth of optical resolution photoacoustic microscopy in tissue is limited to ~ 1mm. It would be helpful if the authors could make a discussion on the impact of their work regarding applications in this field.*

[Reply]: Pursuant to your suggestions, in the revised manuscript, we have added discussions on the potential applications of our work.

“Overall, our work brings a new step change to PAM technologies and will motivate studies in biology and medicine where *in vivo* 3D imaging with spatially invariant resolution is desirable. For instance, SIR-PAM can enable high-resolution functional imaging of angiogenesis, melanoma, and hemoglobin oxygen saturation (sO₂) within the skin and other superficial organs by measuring optical absorption contrasts. Its motionless volumetric imaging ability paves the way to whole-body study of animal embryos *in vivo*. Furthermore, in combination with optical clearing, SIR-PAM can provide deeper imaging. Apart from providing the new perspective for PAM, the DMD-based PISF illumination method may be further exploited for structured-illumination optical microscopy and 3D single-pixel imaging.” (Lines 209 – 216, Page 11)

2.3. *In this manuscript, the authors claimed a resolution-invariant axial range of 1.8mm. What is the corresponding FOV with invariant resolution at +/- 0.9mm depth? This data may help to make*

a fair comparison between the proposed technology and conventional PAM. It is also preferable to include a brief description on the conventional PAM setup used here for comparison.

[Reply]: Another good suggestion. As shown in Fig. 1(a), for SIR-PAM, the invariant-resolution region is determined by the overlap of two collimated beams with the largest incident angle, so it will decrease with the distance from the focal plane. For the current SIR-PAM system, the resolution-invariant region versus the distance from the focal plane can be calculated via $(180 - 0.2|d|) \times (180 - 0.2|d|) \mu\text{m}^2$ with the given NA and illumination area, where d is the distance with a unit of μm . Thus, when the distance d goes up to $\pm 900 \mu\text{m}$, the resolution-invariant region approaches 0.

However, for conventional PAM, the highest lateral resolution can be achieved only at the focal plane, and it suffers rapidly degrading lateral resolution with the distance from the focal plane. Thus there is no resolution-invariant region along the depth direction for conventional PAM.

Accordingly, clarifying language has been added in our revised manuscript: “Besides, due to the limited range of PISFs, the resolution-invariant region in a transverse plane perpendicular to the optical axis decreases with the distance from the focal plane.” (Lines 97 – 99, Page 5).

In addition, a schematic of the conventional PAM system used here, as well as the corresponding brief description, is added to the supplementary material (Supplementary Note 5 and Fig. S4) for comparison, as follows.

“To perform the comparative experiments, we established a conventional PAM system as sketched in Fig. S4. We used a pulsed laser (532 nm wavelength; Elforlight, Ltd.) with a 1-kHz pulse repetition rate as the light source. The pulse energy was monitored by a photodiode detector (SM05PD1A, Thorlabs, Inc.) to compensate for energy fluctuations. The laser beam was expanded and collimated by a beam expander. Instead of being modulated by a DMD, the collimated beam was directly focused by a microscopic objective (NA = 0.1; Olympus, Corp.) to achieve nearly diffraction-limited optical focusing. A PA image was acquired by 2D focal scanning over the entire region of interest, different from the PISF illumination method adopted in SIR-PAM. Here the 2D raster scanning was implemented using two high-resolution translation stages (PLS-85, Physik Instrumente, GmbH & Co. KG). The PA wave was detected by a focused ultrasonic transducer coupled by water, and then amplified by two electronic amplifiers with a combined gain of 48 dB.

A data acquisition system (DAQ, ATS9350, Alazar Technologies, Inc.) was used to simultaneously acquire and digitize the PA signal from the amplifier and the laser intensity signal from the photodiode. Then an image of the optical absorption distribution within the object was formed.”

Figure S4 | Schematic illustration of a conventional PAM system.

2.4. Line 122 on page 7, the authors claimed a resolution-invariant axial range of 1.8mm, within 8% degradation. What is the DOF defined by the 50MHz focused transducer used? It may not affect the resolution of the proposed system which is based on optical resolution. But what is the difference if using an unfocused transducer instead of a focused transducer? How does it affect the proposed system? SNR etc.

[Reply]: Good question. The DOF defined by the 50-MHz focused transducer is 3.8 mm. As you mentioned, the lateral resolution of our proposed method is based on optical resolution, so the transducer’s DOF does not affect the system’s lateral resolution. In our system, a focused transducer is preferable to an unfocused one because the focused transducer enables a higher SNR. Accordingly, we have added these details in our revised manuscript (Line 117, Page 6).

2.5. Line 98 on page 5, the authors derived the FWHM formula, which results in the FWHM of 1.86 μm and 2.71 μm for the proposed system and conventional PAM respectively. However, at line

129 and line 130, the resolution of proposed system and conventional PAM were $1.23\ \mu\text{m}$ and $1.95\ \mu\text{m}$, respectively, less than the calculated value. Please comment on this.

[Reply]: That is really a good question. The lateral resolution defined by the FWHM of the point spread function is the minimum distance between two resolvable points. For the resolution target, the minimum line width we can clearly resolve is not the resolution defined by the point spread function. In the previous version of the manuscript, we claimed that the minimum line widths of the proposed system and conventional PAM were $1.23\ \mu\text{m}$ and $1.95\ \mu\text{m}$, but not the resolution.

For a 1951 USAF resolution target, the definition of the resolution is (https://en.wikipedia.org/wiki/1951_USAF_resolution_test_chart):

$$\text{Resolution (lp/mm)} = 2^{\text{Group}+(\text{element}-1)/6}$$

where the line pair (lp) means a black and white line. In the experiments, the features within the target can be clearly resolved by SIR-PAM and conventional PAM correspond to resolutions of 406.4 lp/mm (group 8, element 5) and 256.0 lp/mm (group 8, element 1), respectively. Accordingly, the line pair widths are actually $2.46\ \mu\text{m}$ and $3.90\ \mu\text{m}$, which are larger than the theoretical predictions.

To avoid ambiguity, in the revised manuscript, we use the resolution of USAF target (lp/mm) instead of minimum line width to judge the performance of the two approaches. (Lines 138 – 139, Page 7).

2.6. Line 165 on page 9, the authors present the laser intensity on the fish surface was $\sim 189\text{mW}/\text{cm}^2$. Usually for the pulsed laser, $20\text{mJ}/\text{cm}^2$ is used as ANSI limit. Is the $\sim 189\text{mW}/\text{cm}^2$ mentioned here based on the peak pulse energy?

[Reply]: Thanks for your query. For conventional PAM that is realized by 2D focal scanning, the PA signal of a point is acquired at a single time, and $20\ \text{mJ}/\text{cm}^2$ per laser pulse is usually used as the ANSI limit. But SIR-PAM employs propagation-invariant sinusoidal fringes to excite the PA wave within the imaged object, so it acquires PA signals corresponding to a certain area at a time with multiple laser pulses. The $20\ \text{mJ}/\text{cm}^2$ per laser pulse limit is automatically satisfied. Thus, we use $200\ \text{mW}/\text{cm}^2$ as the ANSI limit here. The $\sim 189\ \text{mW}/\text{cm}^2$ mentioned here is based on the average pulse energy because of the changed sinusoidal fringes illumination in SIR-PAM. We have mentioned this point in the revised manuscript.

Reviewer #3:

In this paper the authors make use of structured illumination approaches to generate propagation invariant sinusoidal functions for imaging in photoacoustic microscopy. This approach generates a depth-invariant lateral resolution for quite a large imaging depth, thus overcoming one of the drawbacks of traditional PAM. The authors present the basis for the generation of the depth-invariant functions and experimental results both on gel and zebrafish.

Overall, the paper is very interesting and the method shows great potential.

[Reply]: We greatly appreciate your highly positive comments.

3.1 *However, the authors have not discussed a key factor when imaging in-vivo, which is the presence of scattering. How does depth-invariance change when imaging in a slightly scattering medium? The equation the authors propose after Eq(1) includes the transfer function $\exp(ikz)$. This transfer function in the presence of scattering has a real and a complex part, thus attenuating higher frequencies more than lower frequencies, changing the depth-invariance. How is this affected in practice? Is it negligible when dealing with tissues?*

[Reply]: Thanks for your insightful comment. In theory, sinusoidal fringes with higher frequencies attenuate and distort more quickly than those with lower frequencies when propagating in a scattering medium. Thus, with an increase of imaging depth, the high-frequency spectral information would attenuate and cause high-frequency spectral noise. Thus, the resolution gets worse with the increase of imaging depth in a scattering medium.

In the described experiments, slightly scattering media, such as scattering gel or zebrafish embryos, were imaged. The experimental results demonstrated that SIR-PAM enables much better performance in extended DOF and lateral resolution than conventional PAM in slight scattering media. Therefore, we think the influence of scattering on the depth-invariance can be negligible when dealing with slightly scattering media.

3.2 *In this paper the authors present results of two cases which present very little scattering: a scattering gel, as appropriate to establish the resolution of the method, and a 3-day zebrafish embryo, which is one of the most used species in optical microscopy due to its transparency. What is missing is a thorough study of the effect of scattering on this new approach, and how it compares with traditional PAM. It is my feeling that traditional PAM may be less sensitive to scattering than*

SIR-PAM, but I might be mistaken. I believe this article would have a much stronger impact if a study of the effect of scattering is included, instead of the simple gel phantom. I would suggest the authors include equivalent measurements in gels which have different values of scattering, starting at 0 (the one measured) and going up to 10cm^{-1} or nearby, comparing PAM vs SIR-PAM as done so far. This way the true potential of SIR-PAM will be well established and indeed would be a very important contribution to the community. I therefore would be happy to accept this paper after this major revision has been included.

[Reply]: We appreciate your pointing this out. To study the scattering effect on SIR-PAM, we imaged tissue-mimicking phantoms with different scattering coefficients. The experiment details and results have been added in Supplementary Note 7 as follows.

“To study how scattering affects the performance of SIR-PAM, we imaged tissue-mimicking phantoms with different scattering coefficients. In the experiments, intralipid (anisotropy $g = 0.9$) was used as the optical scattering agent, and we mixed it with gelatin-water solution at different concentrations. During this study, the concentrations of these phantoms were set to be 0%, 0.2%, 0.4%, 0.6%, 0.8%, and 1%, which correspond to reduced scattering coefficients μ_s' of = 0, 2, 4, 6, 8 and 10 cm^{-1} , respectively. Two crossed fibers were placed as the imaging targets at a 1-mm depth inside these phantoms. Figure S6 shows the imaging results obtained using SIR-PAM and conventional PAM. It can be seen that conventional PAM can resolve the crossed fibers when μ_s' reaches 10 cm^{-1} , while SIR-PAM can resolve the crossed fibers consistently when μ_s' reaches 6 cm^{-1} and partially when μ_s' reaches 8 cm^{-1} . To quantify the visibility, we show the contrast-to-noise ratios (CNR) of the targets in Fig. S7. If we define the borderline CNR needed to discern the target at a 1-mm depth as 2, the limits of the reduced scattering coefficients for conventional PAM and SIR-PAM are approximately 10 cm^{-1} and $6 - 8\text{ cm}^{-1}$, respectively. Therefore, SIR-PAM is more susceptible to scattering than conventional PAM.”

From the experiments, SIR-PAM is found to be more susceptible to scattering than conventional PAM. Even so, optical clearing can significantly suppress the influence of scattering and help take full advantage of SIR-PAM. We have experimentally demonstrated the enhancement of imaging depth in biological tissue with the help of optical clearing. The details are presented in

Supplementary Note 8 and the results are shown in Fig. S8. Also, we have added a paragraph in the discussion section of our revised manuscript, as follows.

“In the experiments described above, only slightly scattering media, such as scattering gel or zebrafish embryos, were imaged. Strong scattering adversely affects the performance of SIR-PAM. To investigate the effect of scattering on SIR-PAM, we imaged tissue-mimicking phantoms with different scattering coefficients (Supplementary Note 7). SIR-PAM was found to be more susceptible to scattering than conventional PAM. To advance the penetration of SIR-PAM, one could consider optical clearing (Supplementary Note 8), which could be suited for selected applications.” (Lines 203 – 208, Page 11)

Figure S6 | Images of crossed fibers embedded in scattering media acquired by SIR-PAM and conventional PAM. The crossed fibers were buried at a 1-mm depth in scattering media with different reduced scattering coefficients.

Figure S7 | Comparison of the contrast-to-noise ratios for conventional PAM and SIR-PAM with respect to different reduced scattering coefficients.

Figure S8 | Enhancement of imaging depth in chicken breast tissue using SIR-PAM with optical clearing. (a, b) Photographs of 1-mm thick chicken breast tissue with (a) and without (b) optical clearing. (c, d) SIR-PAM images of two crossed carbon fibers beneath 1-mm thick chicken breast tissue with (c) and without (d) optical clearing.

REVIEWERS' COMMENTS:

Reviewer #1 (Remarks to the Author):

The authors have addressed my concerns, and I recommend publication in the journal.

Reviewer #2 (Remarks to the Author):

The authors' response letter and the revised manuscript have clearly addressed the points from the reviewers. New experiment results on the scattering effect on the SIR-PAM have been added as well. Therefore, I am satisfied with the revision made on the manuscript and would like to recommend the manuscript for publication in Nature Communications.

Reviewer #3 (Remarks to the Author):

I believe the authors have done a great job addressing the concerns of the reviewers. I believe that the new revised version of the manuscript is very complete and is ready for publication in Nature Communications.

Point-by-point responses to the reviewers' comments

Reviewer #1:

The authors have addressed my concerns, and I recommend publication in the journal.

[Reply]: We greatly appreciate your comments.

Reviewer #2:

The authors' response letter and the revised manuscript have clearly addressed the points from the reviewers. New experiment results on the scattering effect on the SIR-PAM have been added as well. Therefore, I am satisfied with the revision made on the manuscript and would like to recommend the manuscript for publication in Nature Communications.

[Reply]: We greatly appreciate your comments.

Reviewer #3:

I believe the authors have done a great job addressing the concerns of the reviewers. I believe that the new revised version of the manuscript is very complete and is ready for publication in Nature Communications.

[Reply]: We greatly appreciate your highly positive comments.